# GRADIENT SURGERY FOR MULTI-TASK LEARNING

## ABSTRACT

While deep learning and deep reinforcement learning systems have demonstrated impressive results in domains such as image classification, game playing, and robotic control, data efficiency remains a major challenge, particularly as these algorithms learn individual tasks from scratch. Multi-task learning has emerged as a promising approach for sharing structure across multiple tasks to enable more efficient learning. However, the multi-task setting presents a number of optimization challenges, making it difficult to realize large efficiency gains compared to learning tasks independently. The reasons why multi-task learning is so challenging compared to single task learning are not fully understood. Motivated by the insight that gradient interference causes optimization challenges, we develop a simple and general approach for avoiding interference between gradients from different tasks, by altering the gradients through a technique we refer to as "gradient surgery". We propose a form of gradient surgery that projects the gradient of a task onto the normal plane of the gradient of any other task that has a *conflicting* gradient. On a series of challenging multi-task supervised and multi-task reinforcement learning problems, we find that this approach leads to substantial gains in efficiency and performance. Further, it can be effectively combined with previously-proposed multi-task architectures for enhanced performance in a model-agnostic way.

## 1   INTRODUCTION

While deep learning and deep reinforcement learning (RL) have shown considerable promise in enabling systems to perform complex tasks, the data requirements of current methods make it difficult to learn a breadth of capabilities particularly when all tasks are learned individually from scratch. A natural approach to such multi-task learning problems is to train a single network on all tasks jointly, with the aim of discovering shared structure across the tasks in a way that achieves greater efficiency and performance than solving the tasks individually. However, learning multiple tasks all at once results in a difficult optimization problem, sometimes leading to *worse* overall performance and data efficiency compared to learning tasks individually (Parisotto et al., 2015; Rusu et al., 2016a). These optimization challenges are so prevalent that multiple multi-task RL algorithms have considered using independent training as a subroutine of the algorithm before distilling the independent models into a multi-tasking model (Levine et al., 2016; Parisotto et al., 2015; Rusu et al., 2016a; Ghosh et al., 2017; Teh et al., 2017), producing a multi-task model but losing out on the efficiency gains over independent training. If we could tackle the optimization challenges of multi-task learning effectively, we may be able to actually realize the hypothesized benefits of multi-task learning without the cost in final performance.

While there has been a significant amount of research in multi-task learning (Caruana, 1997; Ruder, 2017), the optimization challenges are not well understood. Prior work has described varying learning speeds of different tasks (Chen et al., 2017) and plateaus in the optimization landscape (Schaul et al., 2019) as potential causes, while a range of other works have focused on the model architecture (Misra et al., 2016b; Liu et al., 2018). In this work, we instead hypothesize that the central optimization issue in multi-task learning arises from gradients from different tasks conflicting with one another. In particular, we define two gradients to be conflicting if they point away from one another (i.e., have a negative cosine similarity). As a concrete example, consider the 2D optimization landscapes of two task objectives shown in Figure 1. The optimization landscape of each task consists of a deep valley, as has been characterized of neural network optimization landscapes in the past (Goodfellow et al., 2014). When considering the combined optimization landscape for multiple tasks, SGD produces gradients that struggle to efficiently find the optimum. This occurs due to a gradient thrashing

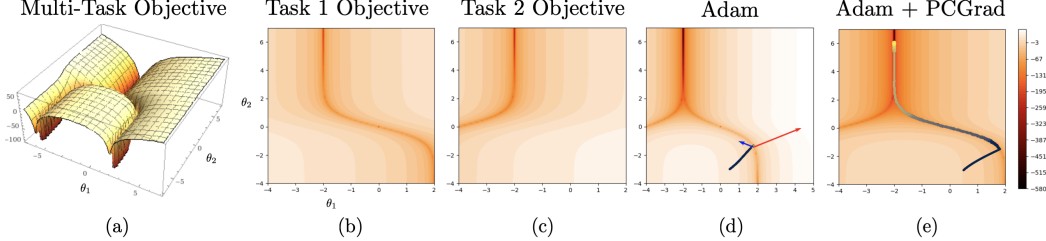

Figure 1: Visualization of PCGrad's effect on a 2D multi-task optimization problem. (a) A multi-task objective landscape. (b) & (c) Contour plots of the individual task objectives that comprise the multi-task objective. (d) Trajectory of gradient updates on the multi-task objective using the Adam optimizer. The gradient vectors of the two tasks at the end of the trajectory are indicated by blue and red arrows, where the relative lengths are on a log scale.(e) Trajectory of gradient updates on the multi-task objective using Adam with PCGrad. For (d) and (e), the optimization trajectory goes from black to yellow.

phenomenon, where the gradient of one task destabilizes optimization in the valley. We can observe this in Figure 1(d) when the optimization reaches the deep valley of task 1, but is prevented from traversing the valley to an optimum. In Section 6.2, we find experimentally that this thrashing phenomenon also occurs in a neural network multi-task learning problem.

The core contribution of this work is a method for mitigating gradient interference by altering the gradients directly, i.e. by performing "gradient surgery". If two gradients are conflicting, we alter the gradients by projecting each onto the normal plane of the other, preventing the interfering components of the gradient from being applied to the network. We refer to this particular form of gradient surgery as *projecting conflicting gradients* (PCGrad). PCGrad is model-agnostic, requiring only a single modification to the application of gradients. Hence, it is easy to apply to a range of problem settings, including multi-task supervised learning and multi-task reinforcement learning, and can also be readily combined with other multi-task learning approaches, such as those that modify the architecture. We evaluate PCGrad on multi-task CIFAR classification, multi-objective scene understanding, a challenging multi-task RL domain, and goal-conditioned RL. Across the board, we find PCGrad leads to significant improvements in terms of data efficiency, optimization speed, and final performance compared to prior approaches. Further, on multi-task supervised learning tasks, PCGrad can be successfully combined with prior state-of-the-art methods for multi-task learning for even greater performance.

## 2 PRELIMINARIES

The goal of multi-task learning is to find parameters $\theta$ of a model $f_\theta$ that achieve high average performance across all the training tasks drawn from a distribution of tasks $p(\mathcal{T})$. More formally, we aim to solve the problem: $\min_\theta \mathbb{E}_{\mathcal{T}_i \sim p(\mathcal{T})} [\mathcal{L}_i(f_\theta)]$, where $\mathcal{L}_i$ is a loss function for the $i$-th task $\mathcal{T}_i$ that we want to minimize. To obtain a model that solves a specific task from the task distribution $p(\mathcal{T})$, we define a task-conditioned model $f_\theta(y|x, z_i)$, with input $x$, output $y$, and encoding $z_i$ for task $\mathcal{T}_i$, which could be provided as a one-hot vector or in any other form.

## 3 MULTI-TASK LEARNING VIA GRADIENT SURGERY

While the multi-task problem can in principle be solved by simply applying a standard single-task algorithm with a suitable task identifier provided to the model or a simple multi-head or multi-output model, a number of prior works (Parisotto et al., 2015; Rusu et al., 2016a; Sener & Koltun, 2018) have found this learning problem to be difficult, especially in the reinforcement learning setting. We hypothesize that one of the main challenges of multi-task learning can be characterized by conflicting and thrashing gradients, and find that this can significantly impede learning progress, especially when combined with iterative data collection. We identify possible causes for this problem and propose a simple and general approach to mitigate it.

### 3.1 THRASHING GRADIENTS IN MULTI-TASK OPTIMIZATION LANDSCAPES

We hypothesize that a key optimization issue in multi-task learning arises when gradients from multiple tasks are in conflict with one another, i.e. when gradients point away from one another. More specifically, we hypothesize that such conflict may lead to *gradient thrashing*. Concretely, gradient

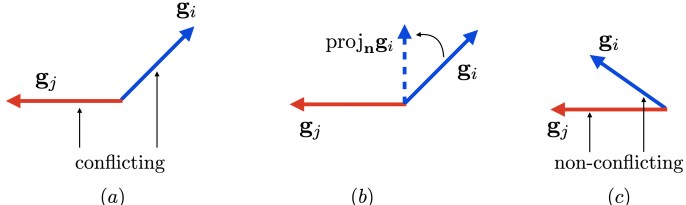

Figure 2: Visual depiction of conflicting gradients and PCGrad. In (a), we see that tasks A and B have conflicting gradient directions, which can lead to destructive interference and unstable learning. In (b), we illustrate the PCGrad algorithm in cases where gradients are conflicting. PCGrad projects the gradient of task A onto the normal vector of task B's gradient. In (c), we show that tasks with non-conflicting gradients are not altered under PCGrad, thereby keeping tasks with constructive interference.

thrashing refers to the phenomenon where a large gradient for one task changes the parameter vectors in a way that substantially decreases performance on another task. Since worse performance typically leads to larger gradients, this results in alternating gradient directions, where, at the next iteration, the second task will have large gradients that dominate and reduce performance on the former task. This issue can be particularly pronounced for neural network optimization, since neural network loss landscapes are known to resemble long narrow valleys (Goodfellow et al., 2014), where the gradient *perpendicular* to the direction of the valley will be small.

We aim to study this hypothesis through two toy examples. First, consider the two-dimensional optimization landscape illustrated in Fig. 1a, where the landscape for each task objective corresponds to a deep and curved valley (Fig. 1b and 1c). The optima of this multi-task objective correspond to where the two valleys meet. More details on the optimization landscape are in Appendix B. We observe that the gradient thrashing hypothesis is consistent with what we observe when running Adam (Kingma & Ba, 2014) on this landscape in Fig. 1d, where we observe that Adam does not traverse one valley towards the other, preventing it from reaching an optimum.

We also aim to detect if a similar phenomenon occurs in multi-task learning with a neural network with thousands of parameters on a toy regression problem. To measure the extent of gradient thrashing, we plot the cosine similarity between the gradients of two tasks throughout the beginning of learning in Fig. 4 (left). We indeed observe a significant level of gradient thrashing at every iteration, where the cosine similarity varies between $-0.75$ and $0.75$ at a very high frequency.

Motivated by these observations, we develop an algorithm that aims to alleviate the optimization challenges caused by gradient thrashing by preventing such gradient conflict between tasks.

### 3.2 PCGRAD: PROJECTING CONFLICTING GRADIENTS

We aim to prevent gradient thrashing by directly altering the gradients themselves, i.e. through "gradient surgery." To be maximally effective and maximally applicable, we must perform surgery in a way that still allows for positive interactions between the task gradients and does not introduce any assumptions on the form of the model.

We start by first detecting whether two gradients are in conflict, by measuring whether they point away from one another. More concretely, we characterize two tasks as conflicting for the current parameter setting if they yield a negative cosine similarity between their respective gradients. The goal of PCGrad is to modify the gradients for each task so as to minimize negative conflict with other task gradients, which will in turn mitigate gradient thrashing.

To deconflict gradients during optimization, PCGrad adopts a simple procedure: if the gradients between two tasks are in conflict, i.e. their cosine similarity is negative, we project the gradient from one task onto the normal plane of the gradient of the other task. This amounts to removing the conflicting component of the gradient for the task, thereby reducing the amount of destructive gradient interference between tasks. A pictorial description of this idea is shown in Fig. 2. Suppose the gradient for task $\mathcal{T}_i$ is $\mathbf{g}_i$, and the gradient for task $\mathcal{T}_j$ is $\mathbf{g}_j$. PCGrad proceeds as follows: (1) First, it determines whether $\mathbf{g}_i$ conflicts with $\mathbf{g}_j$ by computing the cosine similarity between vectors $\mathbf{g}_i$ and $\mathbf{g}_j$, where negative values indicate conflicting gradients. (2) If the cosine similarity is negative, we replace $\mathbf{g}_i$ by its projection onto the normal plane of $\mathbf{g}_j$: $\mathbf{g}_i = \mathbf{g}_i - \frac{\mathbf{g}_i \cdot \mathbf{g}_j}{\|\mathbf{g}_j\|^2} \mathbf{g}_j$. If the gradients are not in conflict, i.e. cosine similarity is non-negative, the original gradient $\mathbf{g}_i$ remains unaltered. (3) PCGrad repeats this process across all of the other tasks sampled in random order from the current

---

**Algorithm 1** PCGrad Update Rule

---

**Require:** Current model parameters $\theta$
  1: Sample mini-batch of tasks $\mathcal{B} = \{\mathcal{T}_k\} \sim p(\mathcal{T})$
  2: **for** $\mathcal{T}_i \sim \mathcal{B}$ in sequence **do**
  3:     Compute gradient $\mathbf{g}_i$ of $\mathcal{T}_i$ as $\mathbf{g}_i = \nabla_\theta \mathcal{L}_i(f_\theta)$
  4:     **for** $\mathcal{T}_j \overset{\text{uniformly}}{\sim} \mathcal{B}$ in random order **do**
  5:         Compute gradient $\mathbf{g}_j$ of task $\mathcal{T}_j$ as $\mathbf{g}_j = \nabla_\theta \mathcal{L}_j(f_\theta)$
  6:         Compute cosine similarity between $\mathbf{g}_i$ as $\mathbf{g}_j$ as $\cos(\phi_{ij}) = \frac{\mathbf{g}_i \cdot \mathbf{g}_j}{\|\mathbf{g}_i\| \|\mathbf{g}_j\|}$.
  7:         **if** $\cos(\phi_{ij}) < 0$ **then**
  8:             Set $\mathbf{g}_i = \mathbf{g}_i - \frac{\mathbf{g}_i \cdot \mathbf{g}_j}{\|\mathbf{g}_j\|^2} \mathbf{g}_j$          // *Subtract the projection of* $\mathbf{g}_i$ *onto* $\mathbf{g}_j$
  9:         **end if**
 10:     **end for**
 11:     Store $\mathbf{g}_i^{\text{proj}} = \mathbf{g}_i$
 12: **end for**
 13: **return** update $\Delta\theta = \sum_i \mathbf{g}_i^{\text{proj}}$

---

batch $\mathcal{T}_j \,\forall\, j \neq i$, resulting in the gradient $\mathbf{g}_i^{\text{proj}}$ that is applied for task $\mathcal{T}_i$. We perform the same procedure for all tasks in the batch to obtain their respective gradients. The full update procedure is described in Algorithm 1 and a discussion on using a random task order is included in Appendix D.

This procedure, while simple to implement, ensures that the gradients that we apply for each task per batch interfere minimally with the other tasks in the batch, mitigating the thrashing gradient problem, producing a variant on standard first-order gradient descent in the multi-objective setting. In practice, the PCGrad gradient surgery method can be combined with any gradient-based optimizer, including commonly used methods such as SGD with momentum and Adam (Kingma & Ba, 2014), by simply passing the computed update to the respective optimizer instead of the original gradient. Our experimental results verify the hypothesis that this procedure reduces the problem of thrashing gradients, and find that, as a result, learning progress is substantially improved.

Finally, we analyze the convergence of this procedure in Theorem 1 in the two-task setting, to ensure that the procedure is sensible under the standard assumptions in optimization.

**Theorem 1.** *Consider two task loss functions $\mathcal{L}_1 : \mathbb{R}^n \to \mathbb{R}$ and $\mathcal{L}_2 : \mathbb{R}^n \to \mathbb{R}$ which are convex and differentiable. For all $\theta \in \mathbb{R}^n$, let $\mathcal{L}(\theta) = \mathcal{L}_1(\theta) + \mathcal{L}_2(\theta)$, i.e. $\mathcal{L}$ is a multi-task objective. Let $\phi$ be the angle between $\nabla \mathcal{L}_1(\theta)$ and $\nabla \mathcal{L}_2(\theta)$. Suppose $\mathcal{L}$ is differentiable and that its gradient is Lipschitz continuous with constant $L > 0$, i.e. we have $\|\nabla \mathcal{L}(\theta_1) - \nabla \mathcal{L}(\theta_2)\|_2 \leq L\|\theta_1 - \theta_2\|_2$ for any $\theta_1, \theta_2$. Then, the PCGrad update rule with step size $t \leq \frac{1}{L}$ will converge to either (1) a location in the optimization landscape where $\cos(\phi) = -1$ or (2) the optimal value $\mathcal{L}(\theta^*)$.*

*Proof.* See Appendix A.                                                                                               □

Theorem 1 states that application of the PCGrad update in the two-task setting with a convex and Lipschitz multi-task loss function $\mathcal{L}$ leads to convergence to either the minimizer of $\mathcal{L}$ or a potentially sub-optimal objective value. A sub-optimal solution occurs when the cosine similarity between the gradients of the two tasks is $-1$, i.e. the gradients directly conflict, leading to zero gradient after applying PCGrad. However, in practice, since we are using SGD, which is a noisy estimate of the true batch gradients, the cosine similarity between the gradients of two tasks in a minibatch is unlikely to be $-1$, thus avoiding this scenario.

## 4   THE PRACTICAL OPERATIONS OF PCGRAD

We apply PCGrad to both supervised learning and reinforcement learning problem settings with multiple tasks or goals. In this section, we discuss the practical instantiations of PCGrad in those settings. Further implementation details are included in Section 6.

### 4.1   MULTI-TASK SUPERVISED LEARNING

In multi-task supervised learning, each task $\mathcal{T}_i \sim p(\mathcal{T})$ has a corresponding training dataset $\mathcal{D}_i$ consisting of $N_i$ labeled training examples, i.e. $\mathcal{D}_i = \{(x, y)_n\}_{n=1}^{N_i}$. The objective for each task

in this supervised setting is then defined as $\mathcal{L}_i(f_\theta) = \mathbb{E}_{(x,y)\sim\mathcal{D}_i} [-\log f_\theta(y|x, z_i)]$, where $z_i$ is a one-hot encoding of task $\mathcal{T}_i$.

At each training step, we randomly sample a batch of data points $\mathcal{B}$ from the whole dataset $\bigcup_i \mathcal{D}_i$ and then group the sampled data with the same task encoding into small batches denoted as $\mathcal{B}_i$ for each $\mathcal{T}_i$ represented in $\mathcal{B}$. We denote the set of tasks appearing in $\mathcal{B}$ as $\mathcal{B}_\mathcal{T}$. After sampling, we precompute the gradient of each task in $\mathcal{B}_\mathcal{T}$ as

$$\nabla_\theta \mathcal{L}_i(f_\theta) = \mathbb{E}_{(x,y)\sim\mathcal{B}_i} [-\nabla_\theta \log f_\theta(y|x, z_i)]. \tag{1}$$

Given the set of precomputed gradients $\nabla_\theta \mathcal{L}_i(f_\theta)$, we also precompute the cosine similarity between all pairs of the gradients in the set. Using the pre-computed gradients and their similarities, we can obtain the PCGrad update by following Algorithm 1, without re-computing task gradients nor backpropagating into the network.

Since the PCGrad procedure is only modifying the gradients of shared parameters in the optimization step, it is model-agnostic and can be readily applied to any architecture designed for supervised multi-task learning. In Section 6, we combine PCGrad with two state-of-the-art architectures for multi-task learning, which leads to noticeable improvement over their original performance.

## 4.2 MULTI-TASK AND GOAL-CONDITIONED REINFORCEMENT LEARNING

For multi-task reinforcement learning, PCGrad can be readily applied to policy gradient methods by directly updating the computed policy gradient of each task, following Algorithm 1, analogous to the supervised learning setting. For actor-critic algorithms, it is also straightforward to apply PCGrad: we simply replace the task gradients for both the actor and the critic by their gradients computed via PCGrad. Hence, PCGrad can be readily incorporated into a variety of model-free RL algorithms. When applying PCGrad to goal-conditioned RL, we represent $p(\mathcal{T})$ as a distribution of goals and let $z_i$ be the encoding of a goal. Similar to the multi-task supervised learning setting discussed above, PCGrad may be combined with various architectures designed for multi-task and goal-conditioned RL (Fernando et al., 2017; Devin et al., 2016), where PCGrad operates on the gradients of shared parameters, leaving task-specific parameters untouched.

In our experiments, we apply PCGrad to the soft actor-critic (SAC) algorithm (Haarnoja et al., 2018), a recently proposed off-policy actor-critic algorithm that has shown significant gains in sample efficiency and asymptotic performance across many different domains. In SAC, we employ a Q-learning style gradient to compute the gradient of the Q-function network, $Q_\phi(s, a, z_i)$, often known as the critic, and a reparameterization-style gradient to compute the gradient of the policy network $\pi_\theta(a|s, z_i)$, often known as the actor. For sampling, we instantiate a set of replay buffers $\{\mathcal{D}_i\}_{\mathcal{T}_i \sim p(\mathcal{T})}$. Training and data collection are alternated throughout training. During a data collection step, we run the policy $\pi_\theta$ on all the tasks $\mathcal{T}_i \sim p(\mathcal{T})$ to collect an equal number of paths for each task and store the paths of each task $\mathcal{T}_i$ into the corresponding replay buffer $\mathcal{D}_i$. At each training step, we sample an equal amount of data from each replay buffer $\mathcal{D}_i$ to form a stratified batch. For each task $\mathcal{T}_i \sim p(\mathcal{T})$, the parameters of the critic $\theta$ are optimized to minimize the soft Bellman residual:

$$J_Q^{(i)}(\phi) = \mathbb{E}_{(s_t,a_t,z_i)\sim\mathcal{D}_i} \left[ Q_\phi(s_t, a_t, z_i) - (r(s_t, a_t, z_i) + \gamma V_{\bar{\phi}}(s_{t+1}, z_i)) \right], \tag{2}$$

$$V_{\bar{\phi}}(s_{t+1}, z_i) = \mathbb{E}_{a_{t+1}\sim\pi_\theta} \left[ Q_{\bar{\phi}}(s_{t+1}, a_{t+1}, z_i) - \alpha \log \pi_\theta(a_{t+1}|s_{t+1}, z_i) \right], \tag{3}$$

where $\gamma$ is the discount factor, $\bar{\phi}$ are the delayed parameters, and $\alpha$ is a learnable temperature that automatically adjusts the weight of the entropy term. For each task $\mathcal{T}_i \sim p(\mathcal{T})$, the parameters of the policy $\pi_\theta$ are trained to minimize the following objective

$$J_\pi^{(i)}(\theta) = \mathbb{E}_{s_t\sim\mathcal{D}_i} \left[ \mathbb{E}_{a_t\sim\pi_\theta(a_t|s_t,z_i))} \left[ \alpha \log \pi_\theta(a_t|s_t, z_i) - Q_\phi(s_t, a_t, z_i) \right] \right]. \tag{4}$$

We compute $\nabla_\phi J_Q^{(i)}(\phi)$ and $\nabla_\theta J_\pi^{(i)}(\theta)$ for all $\mathcal{T}_i \sim p(\mathcal{T})$ and apply PCGrad to both following Algorithm 1.

In the context of SAC specifically, we further study how the temperature $\alpha$ should be adjusted. If we use a single learnable temperature for adjusting entropy of the multi-task policy $\pi_\theta(a|s, z_i)$, SAC may stop exploring once all easier tasks are solved, leading to poor performance on tasks that are

harder or require more exploration. To address this issue, we propose to learn the temperature on a per-task basis, i.e. using a parametrized model to represent $\alpha_\psi(z_i)$ (which we abbreviate as **PA** for per-task alpha). This allows the method to control the entropy of $\pi_\theta(a|s, z_i)$ per-task. We optimize the parameters of $\alpha_\psi(z_i)$ using the same constrained optimization framework as in Haarnoja et al. (2018).

## 5 RELATED WORK

Algorithms for multi-task learning typically consider how to train a single model that can solve a variety of different tasks (Caruana, 1997; Bakker & Heskes, 2003; Ruder, 2017). The multi-task formulation has been applied to many different settings, including supervised learning (Zhang et al., 2014; Long & Wang, 2015; Yang & Hospedales, 2016; Sener & Koltun, 2018; Zamir et al., 2018) and reinforcement-learning (Espeholt et al., 2018; Wilson et al., 2007), as well as many different domains, such as vision (Bilen & Vedaldi, 2016; Misra et al., 2016a; Kokkinos, 2017; Liu et al., 2018; Zamir et al., 2018), language (Collobert & Weston, 2008; Dong et al., 2015; McCann et al., 2018; Radford et al., 2019) and robotics (Riedmiller et al., 2018; Wulfmeier et al., 2019; Hausman et al., 2018). While multi-task learning has the promise of accelerating acquisition of large task repertoires, in practice it presents a challenging optimization problem, which has been tackled in several ways in prior work.

A number of architectural solutions have been proposed to the multi-task learning problem based on multiple modules or paths (Fernando et al., 2017; Devin et al., 2016; Misra et al., 2016b; Rusu et al., 2016b; Rosenbaum et al., 2018; Vandenhende et al., 2019; Rosenbaum et al., 2018), or using attention-based architectures (Liu et al., 2018; Maninis et al., 2019). Our work is agnostic to the model architecture and can be combined with prior architectural approaches in a complementary fashion. A different set of multi-task learning approaches aim to decompose the problem into multiple local problems, often corresponding to each task, that are significantly easier to learn, akin to divide and conquer algorithms (Levine et al., 2016; Rusu et al., 2016a; Parisotto et al., 2015; Teh et al., 2017; Ghosh et al., 2017; Czarnecki et al., 2019). Eventually, the local models are combined into a single, multi-task policy using different distillation techniques (outlined in (Hinton et al., 2015; Czarnecki et al., 2019)). In contrast to these methods, we propose a simple and cogent scheme for multi-task learning that allows us to learn the tasks simultaneously using a single, shared model without the need for network distillation.

Similarly to our work, a number of prior approaches have observed the difficulty of optimization in the multi-task learning setting (Hessel et al., 2019; Chen et al., 2018; Kendall et al., 2018b; Schaul et al., 2019). Our work, in contrast to many of these optimization schemes, suggests that the challenge in multi-task learning may be attributed to the problem of gradient thrashing, which we address directly by introducing a simple and practical algorithm that de-conflicts gradients from different tasks. Prior work (Sener & Koltun, 2018) alternatively proposes a gradient-based multi-objective optimization problem for multi-task learning to address the problem of optimizing possibly conflicting objectives. As noted in Alg 2 in (Sener & Koltun, 2018), it learns a constant scaling factor for per-task gradient to avoid conflicting, while our method corrects both the scaling factor and the direction of per-task gradient, which can more effectively deconflict gradients. Prior work has also used the cosine similarity between gradients to define when an auxiliary task might be useful for single-task learning (Du et al., 2018). We similarly use cosine similarity between gradients to determine if the gradients between a pair of tasks are in conflict. Unlike Du et al. (2018), we use this measure of gradient conflict as a part of gradient surgery in the context of multi-task learning applications.

A number of works in continual learning have studied how to make gradient updates that do not adversely affect other tasks by projecting the gradients into a space that do not conflict with previous tasks (Lopez-Paz & Ranzato, 2017; Chaudhry et al., 2018). Those methods focus on the continual learning setting, and either need to solve for the gradient projections using quadratic programming (Lopez-Paz & Ranzato, 2017), or only projecting the gradient onto the normal plane of the average of the gradients of past tasks (Chaudhry et al., 2018). In contrast, our work focuses on multi-task learning, does not require solving any QP, and *iteratively* projects the gradients of each task onto the normal plane of the gradients of each of the other tasks instead of *averaging*. Finally, our method is distinct from and solves a different problem than the projected gradient method (Calamai & Moré, 1987), which is an approach for constrained optimization that projects gradients onto the constraint manifold.

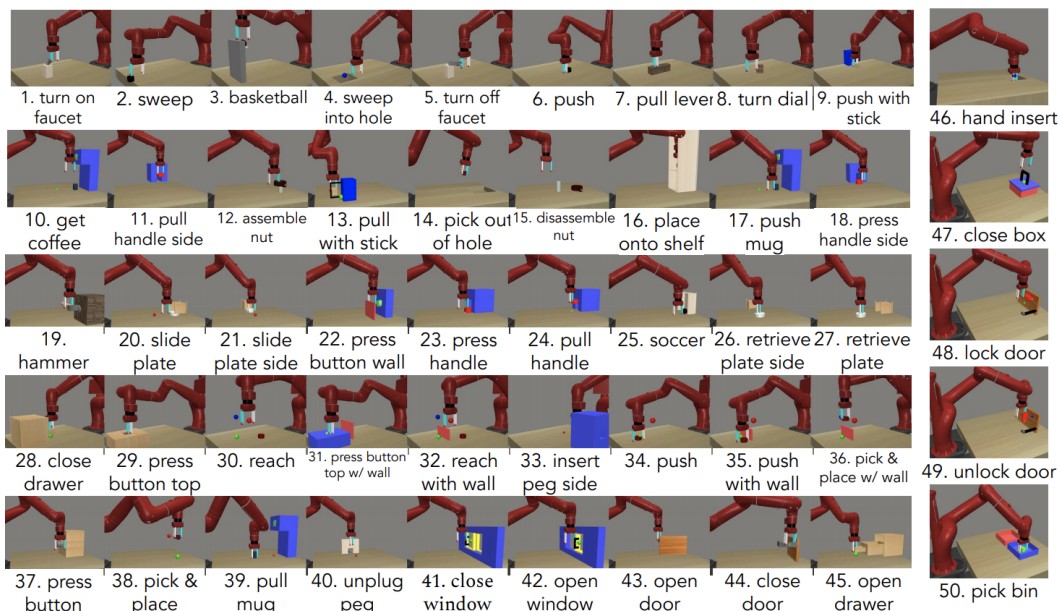

Figure 3: We show visualization of 50 tasks used in MT50 from Meta-World (Yu et al., 2019), which we use for our multi-task RL experiments. MT10 is a subset of the total 50 tasks, which includes reach, push, pick & place, open drawer, close drawer, open door, press button top, open window, close window, and insert peg inside.

# 6 EXPERIMENTS

The goal of our experiments is to study the following questions: (1) Are conflicting gradients a major factor in making optimization for multi-task learning challenging? (2) Does PCGrad make the optimization problems easier for various multi-task learning problems including supervised, reinforcement, and goal-conditioned reinforcement learning settings across different task families? (3) Can PCGrad be combined with other multi-task learning approaches to further improve performance?

## 6.1 EXPERIMENTAL SETUP

To evaluate our method experimentally, we consider both a multi-task supervised learning and a multi-task reinforcement learning problem setup. For supervised learning, we first consider the MultiMNIST dataset (Sener & Koltun, 2018), which contains two tasks: classifying the digit on the top left and on the bottom right in an overlaid image. Beyond digit classification, we also use the CIFAR-100 dataset (Krizhevsky et al., 2009) where each of the 20 label superclasses are treated as distinct tasks, following Rosenbaum et al. (2018). We also conduct experiments on the NYUv2 dataset (Silberman et al., 2012), which consists of RGB-D indoor scene images. Following Liu et al. (2018), we evaluate our method on 3 tasks: 13-class semantic segmentation, depth estimation, and surface normal prediction. In the case of multi-task reinforcement learning, we evaluate our algorithm on the recently proposed Meta-World benchmark (Yu et al., 2019). This benchmark includes a variety of simulated robotic manipulation tasks contained in a shared, table-top environment with a simulated Sawyer arm (visualized as the "Push" environment in Fig. 3). In particular, we use the multi-task benchmarks MT10 and MT50, which consists of the 10 tasks and 50 tasks respectively depicted in Fig. 3 that require diverse strategies to solve them, which makes them difficult to optimize jointly with a single policy. Note that MT10 is a subset of MT50. To evaluate goal-conditioned RL scenarios, we consider goal-conditioned robotic pushing with a Sawyer robot. This domain is representative of challenges in learning goal-conditioned policies over a wide distribution of goals. For details on the experimental set-up and model architectures see Appendix E.

## 6.2 ANALYSIS OF CONFLICTING GRADIENTS

To answer question (1), we consider a simple regression problem, where each task is regressing the input to the output of a sine function. The amplitude and the phase of each task are varied. We construct 10 tasks with the amplitude uniformly sampled from the range $[0, 5]$ and the phase uniformly sampled from the range $[0, \pi]$. The input is also uniformly sampled from the range $[0, 5]$

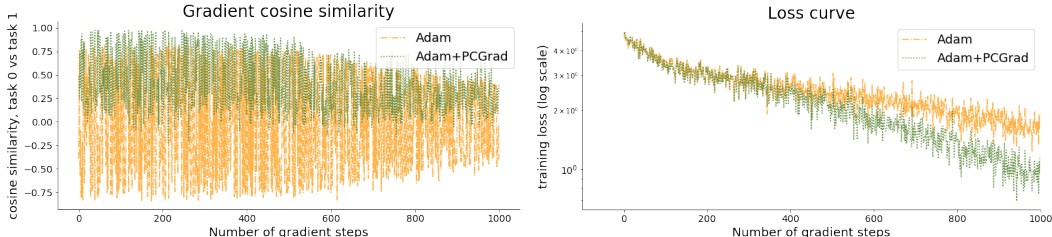

Figure 4: An analysis of the gradients during the first 1000 updates of training, on a toy 10-task sinusoid regression problem. **Left**: The cosine similarity between the gradients of 2 of the 10 tasks (selected arbitrarily, and fixed throughout this plot). We observe a substantial amount of thrashing with standard Adam training, while Adam with PCGrad reduces the thrashing and leads to more closely aligned updates. **Right**: Adam with PCGrad improves performance compared to standard Adam.

and is concatenated with the one-hot task encoding. For training, we use a 3-layer fully-connected neural network with 100 hidden units.

We compare the performance of the network trained with Adam and the network trained with Adam with PCGrad-modified gradients while plotting the cosine similarity between a pair of tasks during training as shown in Figure 4. The plot on the left in Figure 4 demonstrates that the cosine similarity of Adam gradients between a pair of tasks has high variance, which leads to the gradient thrashing problem, while the cosine similarity of the gradient projected by PCGrad yields positive values diminishing the conflicting-gradients problem. As shown in the plot on the right in Figure 4, Adam with PCGrad leads to faster learning over Adam, which implies that gradient thrashing is indeed a problem in multi-task optimization and reducing it can result in considerable performance boost.

## 6.3 MULTI-TASK AND MULTI-OBJECTIVE SUPERVISED LEARNING WITH PCGRAD

To answer question (3), we perform experiments on three standard multi-task supervised learning datasets: MultiMNIST, multi-task CIFAR-100 and NYUv2. We include the results on MultiMNIST in Appendix C.

For CIFAR-100, we follow (Rosenbaum et al., 2018) to treat 20 coarse labels in the dataset as distinct tasks and create a dataset with 20 tasks and 2500 training instances as well as 500 test instances per task. We combine PCGrad with a powerful multi-task learning architecture, routing networks (Rosenbaum et al., 2018; 2019), by simply projecting gradients of the shared parameters in routing networks. As shown in Table 1, applying PCGrad to a single network achieves 71% classification accuracy, which outperforms most of the prior methods such as independent training and cross-stitch (Misra et al., 2016b). Though routing networks achieve better performance than PCGrad on its own, PCGrad is complementary to routing networks and combining PCGrad with routing networks leads to a 2.8% absolute improvement in test accuracy averaged over 3 runs.

We also combine PCGrad with another state-of-art multi-task learning algorithm, MTAN (Liu et al., 2018), and evaluate the performance on a more challenging indoor scene dataset, NYUv2, which contains 3 tasks as described in Section 6.1. We compare MTAN with PCGrad to a list of methods mentioned in Section 6.1, where each method is trained with three different weighting schemes as in (Liu et al., 2018), equal weighting, weight uncertainty (Kendall et al., 2018a), and DWA (Liu et al., 2018). We only run MTAN with PCGrad with weight uncertainty as we find weight uncertainty as the most effective scheme for training MTAN. The results comparing Cross-Stitch, MTAN and MTAN + PCGrad are presented in Table 2 while the full comparison can be found in Table 4 in the Appendix E.3. MTAN with PCGrad is able to achieve the best scores in 8 out of the 9 categories where there are 3 categories per task.

Our multi-task supervised learning results demonstrate that PCGrad can be seamlessly combined with state-of-art multi-task learning architectures and further improve their results on established supervised multi-task learning benchmarks.

## 6.4 MULTI-TASK REINFORCEMENT LEARNING

To answer question (2), we test all methods on 10 and 50 manipulation tasks respectively shown in Figure 3. At each data collection step, we collect 600 samples for each task, and at each training step,

|  | % accuracy |
|---|---|
| task specific-1-fc (Rosenbaum et al., 2018) | 42 |
| task specific-all-fc (Rosenbaum et al., 2018) | 49 |
| cross stitch-all-fc (Misra et al., 2016b) | 53 |
| routing-all-fc + WPL (Rosenbaum et al., 2019) | 74.7 |
| independent | 67.7 |
| PCGrad (ours) | 71 |
| routing-all-fc + WPL + PCGrad (ours) | $\boxed{77.5}$ |

Table 1: CIFAR-100 multi-task results. We apply PCGrad to the routing networks and achieve a significant improvement in classfication accuracy.

| #P. | Architecture | Weighting | Segmentation (Higher Better) | | Depth (Lower Better) | | Surface Normal | | | | |
|---|---|---|---|---|---|---|---|---|---|---|---|
| | | | | | | | Angle Distance (Lower Better) | | Within $t°$ (Higher Better) | | |
| | | | mIoU | Pix Acc | Abs Err | Rel Err | Mean | Median | 11.25 | 22.5 | 30 |
| $\approx 3$ | Cross-Stitch‡ | Equal Weights | 14.71 | 50.23 | 0.6481 | 0.2871 | 33.56 | 28.58 | 20.08 | 40.54 | 51.97 |
| | | Uncert. Weights* | 15.69 | 52.60 | 0.6277 | 0.2702 | 32.69 | 27.26 | 21.63 | 42.84 | 54.45 |
| | | DWA†, $T = 2$ | **16.11** | **53.19** | **0.5922** | **0.2611** | **32.34** | **26.91** | **21.81** | **43.14** | **54.92** |
| 1.77 | MTAN† | Equal Weights | **17.72** | 55.32 | **0.5906** | 0.2577 | 31.44 | **25.37** | $\boxed{23.17}$ | 45.65 | 57.48 |
| | | Uncert. Weights* | 17.67 | **55.61** | 0.5927 | 0.2592 | **31.25** | 25.57 | 22.99 | **45.83** | **57.67** |
| | | DWA†, $T = 2$ | 17.15 | 54.97 | 0.5956 | **0.2569** | 31.60 | 25.46 | 22.48 | 44.86 | 57.24 |
| 1.77 | MTAN† + PCGrad (ours) | Uncert. Weights* | $\boxed{20.17}$ | $\boxed{56.65}$ | $\boxed{0.5904}$ | $\boxed{0.2467}$ | $\boxed{30.01}$ | $\boxed{24.83}$ | 22.28 | $\boxed{46.12}$ | $\boxed{58.77}$ |

Table 2: We present the results on three tasks on the NYUv2 dataset: 13-class semantic segmentation, depth estimation, and surface normal prediction results. #P shows the total number of network parameters. We highlight the best performing combination of multi-task architecture and weighting in bold. The top validation scores for each task are annotated with boxes. The symbols indicate prior methods: *: (Kendall et al., 2018a), †: (Liu et al., 2018), ‡: (Misra et al., 2016b). Performance of other methods as reported in (Liu et al., 2018).

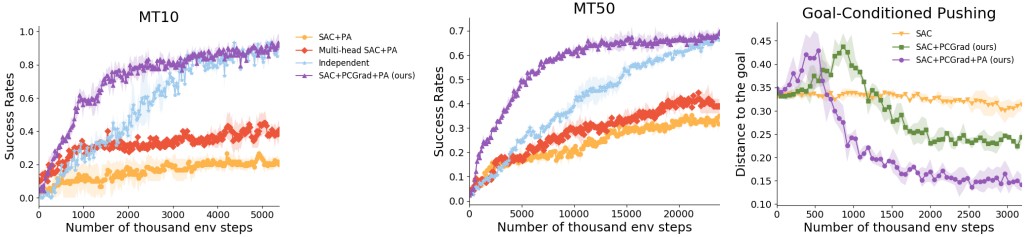

Figure 5: Learning curve on MT10, MT50 and goal-conditioned pushing. PCGrad outperforms the other methods in the three settings in terms of both success rates / average distance to the goal and data efficiency.

we sample 128 datapoints per task from corresponding replay buffers. The results are shown in the two plots on the left in Figure 5. We measure success according to the metrics used in the Meta-World benchmark where the reported the success rates are averaged across tasks. For all methods, we apply **PA** as discussed in Section 4 to learn a separate alpha term per task as the task encoding in MT10 and MT50 is just a one-hot encoding. PCGrad combined with SAC learns all tasks with the best data efficiency and successfully solves all of the 10 tasks in MT10 and about 70% of the 50 tasks in MT50. Training a single SAC policy and a multi-head policy turns out to be unable to acquire half of the skills in both MT10 and MT50, suggesting that eliminating gradient interference across tasks can significantly boost performance of multi-task RL. Training independent SAC agents is able to eventually solve all tasks in MT10 and 70% of the tasks in MT50, but requires about 2 millions and 15 millions more samples than PCGrad with SAC in MT10 and MT50 respectively, implying that applying PCGrad can result in leveraging shared structure among tasks that expedites multi-task learning.

As noted by Yu et al. (2019), these tasks involve fairly distinct behavior motions, which makes learning all of them with a single policy challenging as demonstrated by poor baseline performance. The ability to learn these tasks together opens the door for a number of interesting extensions to meta-learning, goal conditioned RL and generalization to novel task families. We present the results of PCGrad on goal-conditioned RL in the following subsection.

We also provide an ablation study on the importance of correcting the gradient direction and scaling the gradient magnitudes in PCGrad. We construct two variants of PCGrad: (1) only applying the gradient direction corrected with PCGrad while keeping the gradient magnitude unchanged and (2) only applying the gradient magnitude computed by PCGrad while keeping the gradient direction unchanged. As shown in the plot on the left in Figure 6, both variants perform worse than PCGrad and the variant where we only vary the gradient magnitudes is much worse than PCGrad. We also compare PCGrad to a prior method GradNorm (Chen et al., 2018), which scales the magnitude of gradients of all the tasks. As shown in the plot on the right in Figure 6, PCGrad significantly outperforms GradNorm. We also notice that the variant of PCGrad where only the gradient magnitudes change gets comparable results to GradNorm, which suggests that its important to modify both the gradient directions and magnitudes to eliminate interference and achieve good multi-task learning results.

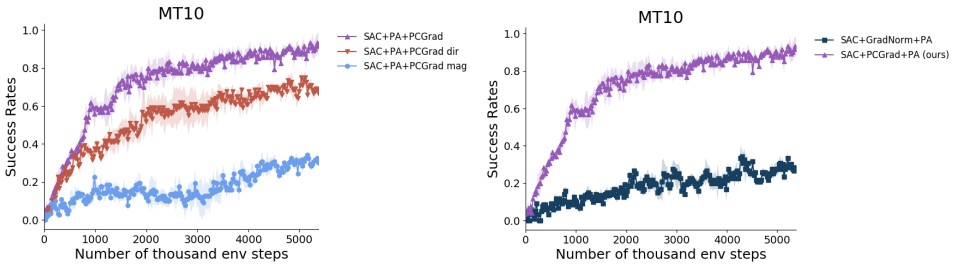

Figure 6: Ablation study on only using the magnitude and the direction of the gradients modified by PCGrad (left) and comparison between PCGrad and GradNorm (Chen et al., 2018) (right). PCGrad outperforms both ablations and GradNorm with a large margin, indicating the importance of modifying both the gradient directions and magnitudes in multi-task learning.

### 6.5 GOAL-CONDITIONED REINFORCEMENT LEARNING

For our goal-conditioned RL evaluation, we use the robot-pushing environment described in Sec. 6.1 where the goals are represented as the concatenations of the initial positions of the puck to be pushed and the its goal location, both of which are uniformly sampled (details in Appendix E.2). We also apply **PA** as discussed in Section 4 to predict the temperature for entropy term given the goal. We summarize the results in the plot on the right in Figure 5. PCGrad with SAC and PA achieves the best performance in terms of average distance to the goal position, while PCGrad with SAC improves over the baseline and a vanilla SAC agent is struggling to successfully accomplish the task. This suggests that PCGrad is able to ease the RL optimization problem also when the task distribution is continuous.

## 7 CONCLUSION

In this work, we identified one of the major challenges in multi-task optimization: conflicting gradients across tasks. We proposed a simple algorithm (PCGrad) to mitigate the challenge of conflicting gradients via "gradient surgery". PCGrad provides a simple way to project gradients to be orthogonal in a multi-task setting, which substantially improves optimization performance, since the task gradients are prevented from negating each other. We provide some simple didactic examples and analysis of how this procedure works in simple settings, and subsequently show significant improvement in optimization for a variety of multi-task supervised learning and reinforcement learning problems. We show that, once some of the optimization challenges of multi-task learning are alleviated by PCGrad, we can obtain the hypothesized benefits in efficiency and asymptotic performance that are believed to be possible in multi-task settings.

While we studied multi-task supervised learning and multi-task reinforcement learning in this work, we suspect the problem of conflicting gradients to be prevalent in a range of other settings and applications, such as meta-learning, continual learning, multi-goal imitation learning (Codevilla et al., 2018), and multi-task problems in natural language processing applications (McCann et al., 2018). Due to its simplicity and model-agnostic nature, we expect that applying PCGrad in these domains to be a promising avenue for future investigation. Further, the general idea of gradient surgery may be an important ingredient for alleviating a broader class of optimization challenges in deep learning, such as the challenges in the stability challenges in two-player games (Roth et al., 2017) and multi-agent optimizations (Nedic & Ozdaglar, 2009). We believe this work to be a step towards simple yet general techniques for addressing some of these challenges.

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

# A  PROOF OF THEOREM 1

*Proof.* We will use the shorthand $|| \cdot ||$ to denote the $L_2$-norm and $\nabla\mathcal{L} = \nabla_\theta\mathcal{L}$, where $\theta$ is the parameter vector. Let $\mathbf{g_1} = \nabla\mathcal{L}_1$, $\mathbf{g_2} = \nabla\mathcal{L}_2$, and $\phi$ be the angle between $\mathbf{g_1}$ and $\mathbf{g_2}$.

At each PCGrad update, we have two cases: $cos(\phi) \geq 0$ or $\cos(\phi < 0)$.

If $\cos(\phi) \geq 0$, then we apply the standard gradient descent update using $t \leq \frac{1}{L}$, which leads to a strict decrease in the objective function value $\mathcal{L}(\phi)$ unless $\nabla\mathcal{L}(\phi) = 0$, which occurs only when $\theta = \theta^*$ (Boyd & Vandenberghe, 2004).

In the case that $\cos(\phi) < 0$, we proceed as follows:

Our assumption that $\nabla\mathcal{L}$ is Lipschitz continuous with constant $L$ implies that $\nabla^2\mathcal{L}(\theta) - LI$ is a negative semidefinite matrix. Using this fact, we can perform a quadratic expansion of $\mathcal{L}$ around $\mathcal{L}(\theta)$ and obtain the following inequality:

$$\mathcal{L}(\theta^+) \leq \mathcal{L}(\theta) + \nabla\mathcal{L}(\theta)^T(\theta^+ - \theta) + \frac{1}{2}\nabla^2\mathcal{L}(\theta)||\theta^+ - \theta||^2$$

$$\leq \mathcal{L}(\theta) + \nabla\mathcal{L}(\theta)^T(\theta^+ - \theta) + \frac{1}{2}L||\theta^+ - \theta||^2$$

Now, we can plug in the PCGrad update by letting $\theta^+ = \theta - t(\nabla\mathcal{L}(\theta) - \frac{\mathbf{g_1}\cdot\mathbf{g_2}}{||\mathbf{g_1}||^2}\mathbf{g_1} - \frac{\mathbf{g_1}\cdot\mathbf{g_2}}{||\mathbf{g_2}||^2}\mathbf{g_2})$. We then get:

$$\mathcal{L}(\theta^+) \leq \mathcal{L}(\theta) + t(\nabla\mathcal{L}(\theta))^T(-\nabla\mathcal{L}(\theta) + \frac{\mathbf{g_1}\cdot\mathbf{g_2}}{||\mathbf{g_1}||^2}\mathbf{g_1} + \frac{\mathbf{g_1}\cdot\mathbf{g_2}}{||\mathbf{g_2}||^2}\mathbf{g_2})$$

$$+ \frac{1}{2}Lt^2||\nabla\mathcal{L}(\theta) - \frac{\mathbf{g_1}\cdot\mathbf{g_2}}{||\mathbf{g_1}||^2}\mathbf{g_1} - \frac{\mathbf{g_1}\cdot\mathbf{g_2}}{||\mathbf{g_2}||^2}\mathbf{g_2}||^2$$

(Expanding, using the identity $\nabla\mathcal{L}(\theta) = \mathbf{g_1} + \mathbf{g_2}$)

$$= \mathcal{L}(\theta) + t(-||\mathbf{g_1}||^2 - ||\mathbf{g_2}||^2 + 2\mathbf{g_1}\cdot\mathbf{g_2} + \frac{(\mathbf{g_1}\cdot\mathbf{g_2})^2}{||\mathbf{g_1}||^2} + \frac{(\mathbf{g_1}\cdot\mathbf{g_2})^2}{||\mathbf{g_2}||^2})$$

$$+ \frac{1}{2}Lt^2||\mathbf{g_1} + \mathbf{g_2} - \frac{\mathbf{g_1}\cdot\mathbf{g_2}}{||\mathbf{g_1}||^2}\mathbf{g_1} - \frac{\mathbf{g_1}\cdot\mathbf{g_2}}{||\mathbf{g_2}||^2}\mathbf{g_2}||^2$$

(Expanding further and re-arranging terms)

$$= \mathcal{L}(\theta) - (t - \frac{1}{2}Lt^2)(||\mathbf{g_1}||^2 + ||\mathbf{g_2}||^2 - \frac{(\mathbf{g_1}\cdot\mathbf{g_2})}{||\mathbf{g_1}||^2} - \frac{(\mathbf{g_1}\cdot\mathbf{g_2})}{||\mathbf{g_2}||^2})$$

$$- Lt^2(\mathbf{g_1}\cdot\mathbf{g_2} - \frac{(\mathbf{g_1}\cdot\mathbf{g_2})^2}{||\mathbf{g_1}||^2||\mathbf{g_2}||^2}\mathbf{g_1}\cdot\mathbf{g_2})$$

(Using the identity $\cos(\phi) = \frac{\mathbf{g_1}\cdot\mathbf{g_2}}{||\mathbf{g_1}||||\mathbf{g_2}||}$)

$$= \mathcal{L}(\theta) - (t - \frac{1}{2}Lt^2)[(1 - \cos^2(\phi))||\mathbf{g_1}||^2 + (1 - \cos^2(\phi))||\mathbf{g_2}||^2]$$

$$- Lt^2(1 - \cos^2(\phi))||\mathbf{g_1}||||\mathbf{g_2}||\cos(\phi)$$

(Note that $\cos(\phi) < 0$ so the final term is non-negative)

Using $t \leq \frac{1}{L}$, we know that $-(1 - \frac{1}{2}Lt) = \frac{1}{2}Lt - 1 \leq \frac{1}{2}L(1/L) - 1 = \frac{-1}{2}$ and $Lt^2 \leq t$.

Plugging this into the last expression above, we can conclude the following:

$$
\mathcal{L}(\theta^+) \leq \mathcal{L}(\theta) - \frac{1}{2}t[(1 - \cos^2(\phi))||\mathbf{g_1}||^2 + (1 - \cos^2(\phi))||\mathbf{g_2}||^2]
$$
$$
- t(1 - \cos^2(\phi))||\mathbf{g_1}||||\mathbf{g_2}|| \cos(\phi)
$$
$$
= \mathcal{L}(\theta) - \frac{1}{2}t(1 - \cos^2(\phi))[||\mathbf{g_1}||^2 + 2||\mathbf{g_1}||||\mathbf{g_2}|| \cos(\phi) + ||\mathbf{g_2}||^2]
$$
$$
= \mathcal{L}(\theta) - \frac{1}{2}t(1 - \cos^2(\phi))[||\mathbf{g_1}||^2 + 2\mathbf{g_1} \cdot \mathbf{g_2} + ||\mathbf{g_2}||^2]
$$
$$
= \mathcal{L}(\theta) - \frac{1}{2}t(1 - \cos^2(\phi))||\mathbf{g_1} + \mathbf{g_2}||^2
$$
$$
= \mathcal{L}(\theta) - \frac{1}{2}t(1 - \cos^2(\phi))||\nabla\mathcal{L}(\theta)||^2
$$

If $\cos(\phi) > -1$, then $\frac{1}{2}t(1 - \cos^2(\phi))||\nabla\mathcal{L}(\theta)||^2$ will always be positive unless $\nabla\mathcal{L}(\theta) = 0$. This inequality implies that the objective function value strictly decreases with each iteration where $\cos(\phi) > -1$.

Hence repeatedly applying PCGrad process can either reach the optimal value $\mathcal{L}(\theta) = \mathcal{L}(\theta^*)$ or $\cos(\phi) = -1$, in which case $\frac{1}{2}t(1 - \cos^2(\phi))||\nabla\mathcal{L}(\theta)||^2 = 0$. Note that this result only holds when we choose $t$ to be small enough, i.e. $t \leq \frac{1}{L}$.

$\square$

## B    2D OPTIMIZATION LANDSCAPE DETAILS

To produce the 2D optimization visualizations in Figure 1, we used a parameter vector $\theta = [\theta_1, \theta_2] \in \mathbb{R}^2$ and the following task loss functions:

$$
\mathcal{L}_1(\theta) = 20 \log(\max(|.5\theta_1 + \tanh(\theta_2)|, 0.000005))
$$
$$
\mathcal{L}_2(\theta) = 25 \log(\max(|.5\theta_1 - \tanh(\theta_2) + 2|, 0.000005))
$$

The multi-task objective is $\mathcal{L}(\theta) = \mathcal{L}_1(\theta) + \mathcal{L}_2(\theta)$. We initialized $\theta = [0.5, -3]$ and performed 500,000 gradient updates to minimize $\mathcal{L}$ using the Adam optimizer with learning rate $0.001$. We compared using Adam for each update to using Adam in conjunction with the PCGrad method presented in Section 3.2.

## C    EXPERIMENTAL RESULTS ON MULTIMNIST

Following the same set-up in  Sener & Koltun (2018), for each image, we sample a different one uniformly at random. Then we put one of the image on the top left and the other one on the bottom right. The two tasks in the multi-task learning problem are to classify the digits on the top left (task-L) and bottom right (task-R) respectively. We construct such 60K examples. We combine PCGrad with the same backbone architecture used in (Sener & Koltun, 2018) and compare its performance to Sener & Koltun (2018) by running the open-sourced code provided in (Sener & Koltun, 2018). As shown in Table 3, our method results 0.13% and 0.55% improvement over Sener & Koltun (2018) in left and right digit accuracy respectively.

|  | left digit | right digit |
|---|---|---|
| Sener & Koltun (2018) | 96.45 | 94.95 |
| PCGrad (ours) | 96.58 | 95.50 |

Table 3: MultiMNIST results. PCGrad achieves improvements over Sener & Koltun (2018) in both left and right digit classfication accuracy.

## D    ABLATION STUDY ON THE TASK ORDER

As stated on line 4 in Algorithm 1, we sample the tasks from the batch and randomly shuffle the order of the tasks before performing the update steps in PCGrad. With random shuffling, we make

PCGrad symmetric w.r.t. the task order in expectation. In Figure 7, we observe that PCGrad with a random task order achieves better performance between PCGrad with a fixed task order in the setting of MT50 where the number of tasks is large and the conflicting gradient phenomenon is much more likely to happen.

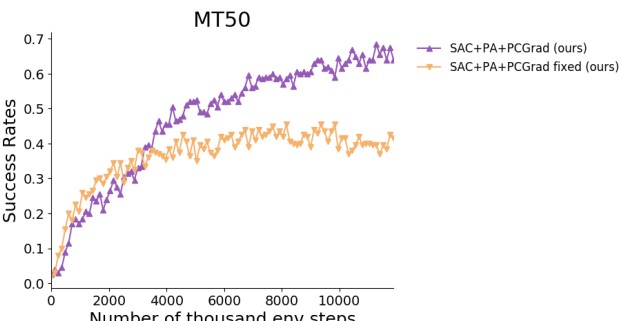

Figure 7: Ablation study on using a fixed task order during PCGrad. PCGrad with a random task order does significantly better PCGrad with a fixed task order in MT50 benchmark.

## E  EXPERIMENT DETAILS

### E.1  DETAILED EXPERIMENT SET-UP

For our CIFAR-100 multi-task experiment, we adopt the architecture used in Rosenbaum et al. (2019), which is a convolutional neural network that consists of 3 convolutional layers with 160 $3 \times 3$ filters each layer and 2 fully connected layers with 320 hidden units. As for experiments on the NYUv2 dataset, we follow Liu et al. (2018) to use SegNet (Badrinarayanan et al., 2017) as the backbone architecture.

Our reinforcement learning experiments all use the SAC (Haarnoja et al., 2018) algorithm as the base algorithm, where the actor and the critic are represented as 6-layer fully-connected feedforward neural networks for all methods. The numbers of hidden units of each layer of the neural networks are 160, 300 and 200 for MT10, MT50 and goal-conditioned RL respectively.

We use five algorithms as baselines in the CIFAR-100 multi-task experiment: **task specific-1-fc** (Rosenbaum et al., 2018): a convolutional neural network shared across tasks except that each task has a separate last fully-connected layer, **task specific-1-fc** (Rosenbaum et al., 2018) : all the convolutional layers shared across tasks with separate fully-connected layers for each task, **cross stitch-all-fc** (Misra et al., 2016b): one convolutional neural network per task along with cross-stitch units to share features across tasks, **routing-all-fc + WPL** (Rosenbaum et al., 2019): a network that employs a trainable router trained with multi-agent RL algorithm (WPL) to select trainable functions for each task, **independent**: training separate neural networks for each task.

For comparisons on the NYUv2 dataset, we consider 5 baselines: **Single Task, One Task**: the vanilla SegNet used for single-task training, **Single Task, STAN** (Liu et al., 2018): the single-task version of MTAN as mentioned below, **Multi-Task, Split, Wide / Deep** (Liu et al., 2018): the standard SegNet shared for all three tasks except that each task has a separate last layer for final task-specific prediction with two variants **Wide** and **Deep** specified in Liu et al. (2018), **Multi-Task Dense**: a shared network followed by separate task-specific networks, **Multi-Task Cross-Stitch** (Misra et al., 2016b): similar to the baseline used in CIFAR-100 experiment but with SegNet as the backbone, **MTAN** (Liu et al., 2018): a shared network with a soft-attention module for each task.

On the multi-task and goal-conditioned RL domain, we apply PCGrad to the vanilla SAC algorithm with task encoding as part of the input to the actor and the critic as described in Section 4 and compare our method to the vanilla **SAC** without PCGrad and training actors and critics for each task individually (**Independent**).

### E.2  GOAL-CONDITIONED EXPERIMENT DETAILS

We use the pushing environment from the Meta-World benchmark (Yu et al., 2019) as shown in Figure 3. In this environment, the table spans from $[-0.4, 0.2]$ to $[0.4, 1.0]$ in the 2D space. To

construct the goals, we sample the intial positions of the puck from the range $[-0.2, 0.6]$ to $[0.2, 0.7]$ on the table and the goal positions from the range $[-0.2, 0.85]$ to $[0.2, 0.95]$ on the table. The goal is represented as a concatenation of the initial puck position and the goal position. Since in the goal-conditioned setting, the task distribution is continuous, we sample a minibatch of 9 goals and 128 samples per goal at each training iteration and also sample 600 samples per goal in the minibatch at each data collection step.

### E.3 FULL NYUv2 RESULTS

We provide the full comparison on the NYUv2 dataset in Table 4.

| Type | #P. | Architecture | Weighting | Segmentation | | Depth | | Surface Normal | | | | |
|---|---|---|---|---|---|---|---|---|---|---|---|---|
| | | | | (Higher Better) | | (Lower Better) | | Angle Distance (Lower Better) | | Within $t°$ (Higher Better) | | |
| | | | | mIoU | Pix Acc | Abs Err | Rel Err | Mean | Median | 11.25 | 22.5 | 30 |
| Single Task | 3 | One Task | n.a. | 15.10 | 51.54 | 0.7508 | 0.3266 | 31.76 | 25.51 | 22.12 | 45.33 | 57.13 |
| | 4.56 | STAN† | n.a. | 15.73 | 52.89 | 0.6935 | 0.2891 | 32.09 | 26.32 | 21.49 | 44.38 | 56.51 |
| Multi Task | 1.75 | Split, Wide | Equal Weights | 15.89 | 51.19 | 0.6494 | 0.2804 | 33.69 | 28.91 | 18.54 | 39.91 | 52.02 |
| | | | Uncert. Weights* | 15.86 | 51.12 | **0.6040** | 0.2570 | **32.33** | **26.62** | **21.68** | **43.59** | **55.36** |
| | | | DWA†, $T = 2$ | **16.92** | **53.72** | 0.6125 | **0.2546** | 32.34 | 27.10 | 20.69 | 42.73 | 54.74 |
| | 2 | Split, Deep | Equal Weights | 13.03 | 41.47 | 0.7836 | 0.3326 | 38.28 | 36.55 | 9.50 | 27.11 | 39.63 |
| | | | Uncert. Weights* | **14.53** | 43.69 | 0.7705 | 0.3340 | **35.14** | **32.13** | **14.69** | **34.52** | **46.94** |
| | | | DWA†, $T = 2$ | 13.63 | **44.41** | **0.7581** | **0.3227** | 36.41 | 34.12 | 12.82 | 31.12 | 43.48 |
| | 4.95 | Dense | Equal Weights | 16.06 | 52.73 | 0.6488 | 0.2871 | 33.58 | 28.01 | 20.07 | 41.50 | 53.35 |
| | | | Uncert. Weights* | **16.48** | **54.40** | 0.6282 | 0.2761 | **31.68** | **25.68** | **21.73** | **44.58** | **56.65** |
| | | | DWA†, $T = 2$ | 16.15 | 54.35 | **0.6059** | **0.2593** | 32.44 | 27.40 | 20.53 | 42.76 | 54.27 |
| | ≈3 | Cross-Stitch‡ | Equal Weights | 14.71 | 50.23 | 0.6481 | 0.2871 | 33.56 | 28.58 | 20.08 | 40.54 | 51.97 |
| | | | Uncert. Weights* | 15.69 | 52.60 | 0.6277 | 0.2702 | 32.69 | 27.26 | 21.63 | 42.84 | 54.45 |
| | | | DWA†, $T = 2$ | **16.11** | **53.19** | **0.5922** | **0.2611** | **32.34** | **26.91** | **21.81** | **43.14** | **54.92** |
| | 1.77 | MTAN† | Equal Weights | **17.72** | 55.32 | **0.5906** | 0.2577 | 31.44 | **25.37** | 23.17 | 45.65 | 57.48 |
| | | | Uncert. Weights* | 17.67 | **55.61** | 0.5927 | 0.2592 | **31.25** | 25.57 | 22.99 | **45.83** | **57.67** |
| | | | DWA†, $T = 2$ | 17.15 | 54.97 | 0.5956 | **0.2569** | 31.60 | 25.46 | 22.48 | 44.86 | 57.24 |
| | 1.77 | MTAN† + PCGrad (ours) | Uncert. Weights* | 20.17 | 56.65 | 0.5904 | 0.2467 | 30.01 | 24.83 | 22.28 | 46.12 | 58.77 |

Table 4: We present the full results on three tasks on the NYUv2 dataset: 13-class semantic segmentation, depth estimation, and surface normal prediction results. #P shows the total number of network parameters. We highlight the best performing combination of multi-task architecture and weighting in bold. The top validation scores for each task are annotated with boxes. The symbols indicate prior methods: *: (Kendall et al., 2018a), †: (Liu et al., 2018), ‡: (Misra et al., 2016b). Performance of other methods taken from (Liu et al., 2018).

