# OpenReview forum: "Gradient Surgery for Multi-Task Learning"
_ICLR.cc/2020/Conference — Reject_

### Official Review · AnonReviewer3 · 2019-10-22
**Official Blind Review #3**

**Rating:** 6

**Review:**

The paper presents a method to boost multi-task learning performance by editing gradient to remove conflicts between tasks. The main idea is to use cosine similarity to 1) determine if two task gradients conflict and 2) to project one conflicting gradient to the normal plane of the other, thereby removing the conflict at the expense of disturbing the other gradient to some extent. Experiments are presented for classification and other computer vision tasks along with reinforcement learning problems.

Overall, I really liked this paper. The explications are clear, the visualizations provided help the understanding (especially Fig.1), and results are compelling. I definitely value the way the method is straightforwardly presented: the underlying idea is simple yet strong. There are, however, a few elements precluding me to pick a higher rating, which I describe in details here.

First, there are a lot of similarities with the MTL method of Sener and Koltun 2018. In particular, I do not agree with the statement that "[this] work, in contrast to many of these optimization schemes [incl. Sener and Koltun], suggests that the challenge in multi-task learning can be attributed to the problem of gradient trashing, which we address directly by introducing a practical algorithm that de-conflicts gradients from different tasks." MTL has the concept of "common descent direction" and, as Fig.1 of relevant paper suggests, it does "de-conflict" the gradients. Sure, the wording is not the same, but the idea is there nonetheless.
To be clear, this is not to say that PCGrad has no merits. I do find it simpler and more elegant (although the latter is a subjective assessment). But I think the similarities should have been discussed in greater details, and the performance compared with Sener and Koltun on at least one problem.

Second, the experiments make it difficult to see the performances of PCGrad alone. Indeed, it is always combined with another multi-task approach/algorithm (MTAN, WPL, SAC+PA, etc.). Providing these results is not incorrect in itself, but it makes it difficult grasp what PCGrad can do alone. Is it worth using only in conjunction with other approaches, or could someone consider using it in a standalone manner? Is PCGrad more a "gradient fine-tuner" than a comprehensive solution for multi-task leearning? The experiments presented, although significant, do not answer these questions.

Third, I am unsure about the value of Theorem 1 and its proof in Sec. 3.2. It assumes too much (e.g., 2 convex tasks) to be of any use in practice. Also, leading to a minimizer of L does not forcibly mean leading to a good solution, depending on the definition of L1 and L2.

There is also one element I'm unsure about. Looking at Algorithm 1, it looks like the order of the tasks may have an effect. Indeed, since it is g_i that is modified at line 8 and not g_j, the last element of B will always remain unaltered (because all other task gradients will have previously been modified to avoid gradient conflict with it). The second to last element of B will be altered, but only due to potential conflicts with the last, and so on, up to the first which can potentially be altered by all others. As an effect, some gradients will always be significantly more altered than others, which can have an impact (at least theoretical) on the learning process. Algorithm 1 thus looks like a greedy approach. Nothing bad in itself (and some ad hoc adjustments, like shuffling B at each update, could very well fix this), but I think this deserves more discussion.

Finally, a more generic comment: the naming of the method should remain the same through the paper. The abstract/intro/Fig.1 refer to the technique as "gradient surgery", while the explanations and experiments talk about PCGrad. Also, in the introduction "plateuas" -> "plateaus".

In summary, I think this is a good paper, presenting a straightforward and useful idea for multi-task learning. However, the related work is not always well described, the experiments lack important comparisons, and the practical effects of PCGrad should be explained in more details instead of focusing on a proof for a convex case.



**Experience Assessment:**

I have read many papers in this area.

**Review Assessment: Checking Correctness Of Derivations And Theory:**

I assessed the sensibility of the derivations and theory.

**Review Assessment: Checking Correctness Of Experiments:**

I carefully checked the experiments.

**Review Assessment: Thoroughness In Paper Reading:**

I read the paper thoroughly.

---

> ### Author Response · Authors · 2019-11-10
> **Author Response**
>
> Thank you for your review! We have addressed all the raised concerns in a revised version of the paper. In the following, we reply to your specific comments.
>
> > “First, there are a lot of similarities with the MTL method of Sener and Koltun 2018...But I think the similarities should have been discussed in greater details, and the performance compared with Sener and Koltun on at least one problem.”
> We revised the paper to include the following discussion: “Sener and Koltun proposes a gradient-based multi-objective optimization problem for multi-task learning to address the problem of optimizing possibly conflicting objectives. As noted in Alg 2 in Sener and Koltun, it learns a scaling factor for per-task gradient to avoid conflicting gradients, while PCGrad corrects both the scaling factor and the direction of per-task gradient, which can more effectively deconflict gradients.”
> We empirically evaluate PCGrad on a MultiMNIST experiment proposed in Sener and Koltun, and find that PCGrad outperforms the method proposed in Sener and Koltun (Table 3). Also note that we added a comparison to GradNorm, as described in the response to R2.
>
> > “Second, the experiments make it difficult to see the performances of PCGrad alone.”
> We compared PCGrad alone to SAC in the goal-conditioned RL experiment (the rightmost plot in Figure 5), where PCGrad alone outperformed SAC. We will add a comparison to PCGrad alone on CIFAR in the next revision.
>
> > “Third, I am unsure about the value of Theorem 1 and its proof in Sec. 3.2. It assumes too much (e.g., 2 convex tasks) to be of any use in practice.”
> We believe the value of the Theorem is as a sanity check that the algorithm is convergent under simplifying conditions and agree that it is not otherwise useful. Convexity is a standard assumption used to theoretically analyze optimization algorithms, including optimizers that are empirically effective with deep networks [1].
>
> >”Algorithm 1 thus looks like a greedy approach. Nothing bad in itself (and some ad hoc adjustments, like shuffling B at each update, could very well fix this), but I think this deserves more discussion.”
> In Alg 1,  we do sample a batch of tasks uniformly from the task distribution, i.e. equivalent to shuffling B as described. We have clarified this in Alg 1 in the revised version.
>
> >”a more generic comment: the naming of the method should remain the same through the paper.”
> We have fixed this issue in the revised version, using only the name PCGrad.
>
> >”the practical effects of PCGrad should be explained in more details”
> We added a subsection (Appendix D) in the paper to discuss the practical effects of PCGrad, where we discuss the effect of the order of the tasks in the batch in Alg 1 and also include an ablation study (Figure 7) that shows shuffled task order leads to better and more stable performance in the multi-task RL setting.
>
> [1] Kingma, D. P., & Ba, J. (2014). Adam: A method for stochastic optimization. arXiv preprint arXiv:1412.6980.

---

> ### Author Response · Authors · 2019-11-15
> **Comparison to PCGrad alone on multi-task CIFAR**
>
> Dear Reviewer 3,
>
> We have added the comparison to PCGrad alone on multi-task CIFAR (see Table 1). PCGrad alone (i.e. applying PCGrad to a single network) achieves 71% classification accuracy, which outperforms other methods such as cross-stitch and independent training. PCGrad alone performs worse than routing networks (74.7%). However, we find that PCGrad is complementary to routing networks -- the combination achieves 77.5% accuracy.
>
> Note that these numbers are all higher than the numbers in the initial submission. This is because we discovered and fixed a bug arising from the open-source routing networks code. The bug was due to all models being trained for only 3 epochs instead of 50. The updated results above and in the paper all train for the full 50 epochs.

---

### Official Review · AnonReviewer1 · 2019-10-22
**Official Blind Review #1**

**Rating:** 3

**Review:**

The paper proposes a simple rule to ignore the so-called conflicting gradients in addressing multi-task learning problems. The underlying idea is very straight-forward if two gradients are contradictory (the angle between them is > \pi) then one should not be considered.

I have some questions here. Let's assume the gradients passing to PCGrad are g1, g2, and g3 (in that order).

1- In PCGrad, we start with g1 and will keep g2, or g3 if their direction complies with g1, isn't it better to keep the gradient that minimizes the loss more (or say has a bigger norm) instead? Maybe one should first pick which gradient is more important and then use that to start PCGrad.

2- One may argue that if <g1,g2> <= 0, <g1,g3> <= 0, PCGrad will ignore both g2 and g3, hence, intuitively it should converge slowly and probably not generalizable. Can you comment on why I should not worry about this?

3- Is it possible to generalize PCGrad to work with more than two gradient vectors? One imagines that if it can work with more than 2 gradients, maybe a better and more robust algorithm can be achieved.

Based on the above, I believe that the paper in its current form has not completely studied the problem and hence I am giving the paper a weak reject score at this stage.


**Experience Assessment:**

I have read many papers in this area.

**Review Assessment: Checking Correctness Of Derivations And Theory:**

I carefully checked the derivations and theory.

**Review Assessment: Checking Correctness Of Experiments:**

I assessed the sensibility of the experiments.

**Review Assessment: Thoroughness In Paper Reading:**

I read the paper thoroughly.

---

> ### Author Response · Authors · 2019-11-07
> **Clarification and Response**
>
> Thank you for your review! We want to clarify a potential misunderstanding of the algorithm. We believe the three main questions/concerns listed in the review stem from this misunderstanding:
> “The underlying idea is very straight-forward if two gradients are contradictory (the angle between them is > \pi) then one should not be considered.”
> This statement is not accurate, which we hope to clarify:
> • Both of the two gradients are updated, rather than just one (Alg 1, Line 2 iterates over the all gradients being updated).
> • Gradients are projected according to the equation stated on line 8 in Alg 1, instead of being “not considered”.
> • The method projects the gradients when the angle between gradients is greater than \pi/2, rather than \pi.
>
> In light of these clarifications, here are responses to each question
> Q1: We update the gradient for all tasks (Alg 1, Line 2), and randomize the projection order for each task gradient (which we will clarify in Alg 1, Line 4). As a result, PCGrad is symmetric to task order in expectation.
> Q2: Because we project and rather than “not considering” the gradient, the gradient will not be ignored.
> Q3: Alg 1 generalizes PCGrad to more than 2 tasks, and all of the results in Sec 6 show results with more than 2 tasks.
>
> Please let us know if you have any comments, questions, or concerns in light of this clarification. Thank you!

---

> ### Author Response · Authors · 2019-11-14
> **Author followup**
>
> Dear Reviewer 1,
>
> Could you let us know if our clarification and response have addressed the concerns raised in your review? We would be happy to provide further revisions or experiments to address any remaining issues, and would appreciate a response from you on the points that we raised.

---

### Official Review · AnonReviewer2 · 2019-10-25
**Official Blind Review #2**

**Rating:** 3

**Review:**

This paper proposes as solution to manage the case where gradients are conflicting in gradient-based Multi-Task Learning (MTL), pointing to different directions. They propose a simple “gradient surgery” technique that alters the gradients by projecting a conflicting gradient on the normal vector of the other one, in order to mitigate the effect. The method is generic in the sense that it can be directly applied to various gradient-based architectures easily.

The paper is well written and easy to follow. However, the whole proposal relies on the assumption that conflicting gradients are common and harmful for MTL. For simple convex models, like the model used in the theorem, I get that this can be an issue. But using a convex model for MTL seems no that common. And as far as I know, MTL is not used commonly with simple models, it is rather common is with deep neural networks, where a common part of the network (i.e. representation) is shared among the tasks, while we have a distinct head, of one or few layers, for each task. With such a setting, we expect to have enough capacity to model the various tasks, such that the neural network model should be able to model both tasks independently if they are in complete contradiction. In such case, over the training, conflicting gradients over some neurons with be a transient phenomenon, with the neurons specializing on one or the other task, or just be disabled. In practice, common elements will be shared, while contradictory elements will be in the task-specific part of the network. Said otherwise, I think that conflicting gradients may appear on some neurons over some data, but will be mostly a stochastic phenomenon which is not necessarily harmful on the long run, much like the stochasticity of picking a sequence of data from different classes in SGD.

I may be not totally right in speculating in such way on what is going on with conflicting gradients in MTL. My point is to show that the whole paper is based on assumptions that are not verified. I would need to be convinced that we are tackling a real problem, not such an idea of something that may happen but in practice is quite rare or not necessarily that harmful. The experiments over the toy problem on trashing gradient (Sec. 3.1 and Fig. 1) is interesting to illustrate the problem, but show the issue in a 2D setting, where the model has very little degree of freedom. With deep networks, I don’t think that such constrained search space is common.

Another issue with the paper is the lack of comparison with other approaches for MTL. A conjecture I have on the performance of the approach is that it may dampen the gradient for the loss of the different tasks, and their amplitude, to get to an effect similar to what GradNorm is doing. In fact, all results reported are on a base method and one with PCGrad. Comparison with other methods to handle MTL is required in my opinion, in particular with GradNorm, which may have similar effects than the current one. If PCGrad and GradNorm achieve similar results, my guess is that the issue is not with conflicting gradients, but rather bad scaling of the losses when they are put together.

** Update ** : After reading the authors' comments and other reviews, I would maintain a "weak reject", although if there was a "borderline" choice, I would have used it. The authors provide reasonable answer to my request, although I think the paper could have been stronger in term of experimentations. Also, the other reviews and comments on existing work make it clear to me that the positioning to other related work was incomplete. I think the proposal and paper is correct and interesting, so still below the acceptance threshold for ICLR.

**Experience Assessment:**

I have published one or two papers in this area.

**Review Assessment: Checking Correctness Of Derivations And Theory:**

I assessed the sensibility of the derivations and theory.

**Review Assessment: Checking Correctness Of Experiments:**

I carefully checked the experiments.

**Review Assessment: Thoroughness In Paper Reading:**

I read the paper at least twice and used my best judgement in assessing the paper.

---

> ### Author Response · Authors · 2019-11-10
> **Author Response**
>
> Thank you for your review! We have addressed all the raised concerns in a revised version of the paper. In the following, we reply to your specific comments.
>
> > “Comparison with other methods to handle MTL is required in my opinion, in particular with GradNorm, which may have similar effects than the current one”.
> We added comparisons to GradNorm in MT10 (see Figure 6) where PCGrad is able to achieve near 100% success rate after 5M samples while GradNorm achieves around 30% success rate.
> PCGrad is scaling the gradient magnitudes and correcting the gradient directions, while GradNorm is only scaling the gradient magnitudes, which may be why PCGrad excels. To verify this hypothesis, we added two ablations of PCGrad: (1) only applying the gradient direction modification of PCGrad while keeping the gradient magnitude unchanged and (2) only applying the gradient magnitude computed by PCGrad while keeping the gradient direction unchanged. As shown in Figure 6, both variants perform worse than PCGrad. Further, the variant where we only vary the gradient magnitudes is much worse than PCGrad but gets comparable results to GradNorm, which suggests that it’s important to modify both the gradient directions and magnitudes to eliminate interference and achieve good multi-task learning results.
>
> > “And as far as I know, MTL is not used commonly with simple models, it is rather common is with deep neural networks...With such a setting, we expect to have enough capacity to model the various tasks, such that the neural network model should be able to model both tasks independently if they are in complete contradiction...I would need to be convinced that we are tackling a real problem, not such an idea of something that may happen but in practice is quite rare or not necessarily that harmful. The experiments over the toy problem on trashing gradient (Sec. 3.1 and Fig. 1) is interesting to illustrate the problem, but show the issue in a 2D setting, where the model has very little degree of freedom.”.
> We agree that MTL is used with deep neural networks, and we studied precisely this setting in our experiments. Specifically, in the left plot in Figure 4, we observed that in the sinusoid regression problem, learning a single model with shared parameters for all the tasks have cosine similarities between task gradients with high variance as well as negative values while cosine similarity of gradients projected by PCGrad produce positive values. Furthermore, in the right plot in Figure 4, learning a single model is performing much worse than PCGrad, implying that over-parameterized neural networks also exhibited the conflicting gradients phenomenon and PCGrad can reduce such a problem to yield better performance. Moreover, PCGrad strongly outperforms prior methods on a range of challenging, real multi-task learning problems (Table 1, Figure 5, Table 4) with deep neural networks, suggesting that, while the models have the capacity to represent all tasks, optimization challenges hinder their ability to realize that capacity and PCGrad is equipped with the ability to mitigate such challenges.

---

### Public Comment · ~Lucas_Deecke1 · 2019-10-31
**Related resources**

After going through the paper, it seems that two related resources have been overlooked.

1.) Lopez-Paz & Ranzato (NeurIPS, 2017) take an optimization-based perspective to motivate the projection of gradients. Their resulting approach, gradient episodic memory (GEM), draws on the same methodical insight discussed here (i.e. they seek alignment between gradients), and is one of the most popular current techniques in the continual learning literature.

2.) Follow-up work by Chaudry et al. (ICLR, 2019) relaxes the projection in GEM (where a sequence of gradients is in involved in every update). As you’ll see from comparing to eq. (11) in their paper, the gradient update they propose is equivalent to the one you use here:
$$g \leftarrow g - \frac{\langle g , g' \rangle }{\langle g' , g' \rangle }g'.$$
Interesting read, and good to see the idea of gradient projections evaluated in a multi-task setting! Nonetheless would be good if the commonalities with previous work were discussed.

Lopez-Paz, David, and Marc'Aurelio Ranzato. "Gradient episodic memory for continual learning.", NeurIPS 2017.
Chaudhry, Arslan, et al. "Efficient lifelong learning with A-GEM.”, ICLR 2019.

---

> ### Author Response · Authors · 2019-11-02
> **Added discussion and comparison**
>
> Hi Lucas,
>
> Thank you for your comment! We did recognize that GEM and A-GEM are relevant to our work after submitting the paper and will include a discussion on those two methods in a revised version of this paper. We will also provide the discussion in the paragraph below. We also added an experiment that compares to an approach analogous to A-GEM in the multi-task setting, and we observe that PCGrad performs substantially better.
>
> A number of works in continual learning have studied how to make gradient updates that do not adversely affect other tasks by projecting the gradients into a space that do not conflict with previous tasks (Lopez-Paz & Ranzato, 2017; Chaudhry et al.,2018). Those methods focus on the continual learning setting, and either need to solve for the gradient projections using quadratic programming (Lopez-Paz & Ranzato, 2017) or only projecting the gradient onto the normal plane of the average of the gradients of past tasks (Chaudhry et al.,2018). In contrast, our work focuses on multi-task learning, does not require solving any QP, and $\textit{iteratively}$ projects the gradients of each task onto the normal plane of the gradients of each of the other tasks, which we find performs much better than $\textit{averaging}$. On the MT10 evaluation mode in Meta-World benchmarks, we find that our iterative projection method manages to solve all of the 10 tasks as seen in Figure 5 in the paper while averaging can only acquire 4 out of the 10 tasks.
>
> Lopez-Paz, David, and Marc’Aurelio Ranzato. “Gradient episodic memory for continual learning.“, NeurIPS 2017.
> Chaudhry, Arslan, et al. “Efficient lifelong learning with A-GEM.“, ICLR 2019.

---

### Decision · Program_Chairs · 2019-12-19

**Decision:**

Reject

**Comment:**

This paper presents a method for improving optimization in multi-task learning settings by minimizing the interference of gradients belonging to different tasks.

While the idea is simple and well-motivated, the reviewers felt that the problem is still not studied adequately. The proofs are useful, but there is still a gap when it comes to practicality.

The rebuttal clarified some of the concerns, but still there is a feeling that (a) the main assumptions for the method need to be demonstrated in a more convincing way, e.g. by boosting the experiments as suggested with other MTL methods (b) by placing the paper better in the current literature and minimizing the gap between proofs/underlying assumptions and practical usefulness.